# Thelytokous Reproduction of Onion Thrips, *Thrips tabaci* Lindeman 1889, Infesting Welsh Onion and Genetic Variation among Their Subpopulations

**DOI:** 10.3390/insects13010078

**Published:** 2022-01-11

**Authors:** Falguni Khan, Miltan Chandra Roy, Yonggyun Kim

**Affiliations:** Department of Plant Medicals, Andong National University, Andong 36729, Korea; falgunikhan2942@gmail.com (F.K.); miltan.roy@yahoo.com (M.C.R.)

**Keywords:** thelytoky, COI, RAPD, thrips population, Welsh onion, Korea

## Abstract

**Simple Summary:**

Parthenogenesis is an asexual type of reproduction that usually occurs in thrips. Thelytokous parthenogenesis is a kind of reproduction that produces female progeny without mating. This study reports a thelytokous reproduction of the onion thrips, *Thrips tabaci* Lindeman 1889, host strain infesting Welsh onion. Cytochrome oxidase I (COI) sequences of the populations exhibited specific residues at conserved positions of thelytokous biotype (called ‘L2’). Phylogenetic tree analysis revealed that COI sequences of the onion thrips collected from different local populations infesting Welsh onion were clustered with L2 biotype populations. In the laboratory, the thelytokous reproduction was demonstrated because each single thrips produced only female progeny. Interestingly, these thelytokous populations collected from different localities showed a certain level of genetic diversity. However, the genetic distance was independent of the actual distance among different local populations. Results of this study indicate that *T. tabaci* infesting Welsh onion is a thelytokous biotype with genetic variation among local populations.

**Abstract:**

Parthenogenesis is not uncommon in thrips. This asexual reproduction produces males (arrhenotokous) or female (thelytokous). Only females are found in the onion thrips (*Thrips tabaci* Lindeman 1889) infesting Welsh onion (*Allium fistulosum*) in several areas of Korea. To determine the reproduction mode of *T*. *tabaci*, thrips infesting Welsh onion were collected from different localities in Korea. Cytochrome oxidase I (COI) sequences were then assessed. Results showed that all test local populations had signature motif specific to a thelytokous type. These COI sequences were clustered with other thelytokous populations separated from arrhenotokous *T*. *tabaci* populations. In a laboratory test, individual rearing produced female progeny without any males. These results support that Korean onion thrips infesting Welsh onion have the thelytokous type of parthenogenesis. Local thrips populations exhibited significant variations in susceptibility to chemical and biological insecticides. Random amplified polymorphic DNA (RAPD) analysis indicated genetic variations of local populations. However, the genetic distance estimated from RAPD was independent of the actual distance among different local populations. These results suggest that genetic variations of *T*. *tabaci* are arisen from population subdivision due to asexual thelytokous reproductive mode.

## 1. Introduction

The onion thrips (*Thrips tabaci* Lindeman 1889) (Thysanoptera: Thripidae) is an important insect pest of onions including Welsh onion (*Allium fistulosum* L.). Its larvae and adults feed on leaf tissues and remove chlorophyll, thus impairing photosynthesis [1]. In addition to directly feeding on plant tissues, the onion thrips play an important role in the transmission of several onion pathogens such as *Pantoea agglomerans* and *P*. *ananatis* known to cause bacterial stalk and leaf necrosis [2,3]. Onion thrips is also known to transmit iris yellow spot virus (IYSV) to onions [4] and tospovirus to mostly solanaceous host plants [5].

Onion thrips is a cosmopolitan species with a wide host range, allowing it to have genetic diversity due to different hosts in various local regions [6]. For example, to control the thrips, foliar spray of chemical insecticides including spinetoram resulted in both regional and temporal variations in its efficacy in USA [7]. This suggests a genetic heterogeneity among the onion thrips populations although they can produce progeny in a parthenogenetic mode from a single parent. The ‘species complex’ issue of *T*. *tabaci* was initially raised by Zawirska [8], who suggested two distinct biotypes within the thrips populations. Currently, *T*. *tabaci* is divided into the following three lineages based on mitochondrial DNA sequences: a tobacco-associated (T) biotype and two leek-associated (L1, L2) biotypes [9,10]. Although all biotypes are parthenogenous, T and L1 biotypes reproduce through arrhenotoky (unfertilized eggs develop to males) while L2 biotype reproduces through thelytoky (unfertilized eggs develop into females) [11].

Welsh onion is cultivated in most agricultural farming areas in Korea, where *T*. *tabaci* causes serious direct feeding damage [12]. A flight behavior analysis has shown that *T*. *tabaci* performs both dispersal and migratory displacement, leading to its wide habitat range in Korea and other countries [13]. However, the biotype of thrips populations infesting the Welsh onion was unknown. In addition, parthenogenesis of thrips might prevent chromosomal recombination, which limits gene diversity of *T*. *tabaci*. This narrow genetic variability might be detrimental to species. However, little is known about the Welsh onion populations of *T*. *tabaci*. Thus, the objective of this study was to determine the biotype of the Welsh onion populations and their genetic diversity. To test genetic diversity, different local populations of *T. tabaci* were assessed in their variation in insecticide susceptibility.

## 2. Materials and Methods

### 2.1. Insect Rearing

Larvae and adults of *T*. *tabaci* were collected from six different regions (See Results). Rearing conditions were temperature, 25 ± 1 °C; photoperiod, 16 h/8 h of light/dark; and relative humidity, 70 ± 5% according to Reiter et al. [14]. Fresh Welsh onion leaves were used to feed onion thrips. Adults laid eggs on onion leaves. Old leaves were transferred to egg boxes to obtain new larvae. When larvae emerged, they were transferred to larvae boxes from egg boxes. In the same way, pupae and adults were transferred to their respective boxes. Circular breeding containers (100 × 40 mm, SPL, Pocheon, Korea) were used to rearing onion thrips.

### 2.2. Field Monitoring of T. tabaci from Three Localities

Three fields (Songcheon, Yongsang 1, and Yongsang 2) of Welsh onion were monitored in this study. Songcheon field cultured onions in greenhouses. Both Yongsang sites were open fields. However, Yongsang 1 was close to a greenhouse culturing Welsh onion. Soon after transplanting onions from the nursery to fields in early May, yellow sticky traps (15 × 25 cm, Green Agrotech, Gyeongsan, Korea) were installed with three replications in each field because sticky traps and cards are used in both open fields and greenhouses for monitoring the early occurrence and change in population size during different seasons [15]. Thrips populations were monitored every week until harvest in August. For the identification of *T*. *tabaci* caught by sticky traps, morphological characters (such as body color yellow or brown or yellowish brown, antennae seven-segmented, well-developed wings, and ovipositor at the tip of the abdomen) were used. To avoid any errors in the identification by contamination of *Frankliniella occidentalis* and *F. intonsa*, which are common in Korea, we used the morphological characters [16] to diagnose these two species.

### 2.3. Laboratory Test of Thelytokous Reproduction

L1 larvae were randomly selected and individually reared in small Petri dishes (40 × 20 mm, SPL) until adult stage. For reproduction tests, adults were kept individually or in a group. Single parent test used 30 individuals (=replications). For four or ten parental tests, reproduction tests were replicated 10 times or 5 times, respectively. Test arena used small Petri dishes with fresh Welsh onion. Progeny larvae were counted at five days after the reproduction test. Adult females were counted by observing ovipositor at 10 days after the reproduction test.

### 2.4. Genomic DNA Extraction

For cytochrome oxidase I (COI) sequencing, genomic DNA (gDNA) was extracted from 5~6 adult thrips which was collected from field population of different regions according to the method described by Kim et al. [16]. Seven different places (see Results) include Seoul (37°36′12″ N, 127°8′37″ E), Suanbo (36°50′54″ N, 127°59′28″ E), Andong (36°33′33″ N, 128°43′44″ E), Yeongdeok (36°24′36″ N, 129°22′30″ E), Gampo (35°80′28″ N, 129°50′28″ E), Yangsan (35°41′46″ N, 129°05′90″ E), and Namwon (35°40′57″ N, 127°36′61″ E). Briefly, each thrips was crushed with a pestle in 30 μL of 20% Chelex (Bio-Rad, Hercules, CA, USA) and heated at 100 °C for 10 min. After cooling on ice for 2 min, the suspension was centrifuged at 14,000× *g* for 5 min. The resulting supernatant (1–2 ng DNA/μL) was used as gDNA sample. 

### 2.5. PCR of Mitochondrial COI

Amplification of the 5′-fragment (482 bp) of cytochrome oxidase 1 (CO1) gene was carried out by PCR with a pair of primers: C1-J-1718 5′-GGAGGATTTGGAAATTGATTAGTTCC-3′ and C1-N-2191 5′-CCCCGGTAAAATTAAAATATAAACTTC-3′ [17]. A reaction mixture (25 μL) consisted of 2 μL of gDNA, 2 mM MgCl_2_, 0.2 mM dNTP, 4 pmol of each primer, and 1 unit of Taq DNA polymerase (GeneAll, Seoul, Korea). PCR was performed on a MyCycler Personal Thermal Cycler (Bio-Rad) using the following cycling conditions: initial heating at 94 °C for 3 min; 35 cycles of denaturation at 94 °C for 1 min, annealing at 50 °C for 1 min, and polymerization at 72 °C for 10 min; followed by keeping at 4 °C. PCR products were cloned into a PCR2.1 cloning vector (Invitrogen, Carlsbad, CA, USA) and transformed into *Escherichia coli* TOP 10 chemical competent cells. Plasmids were obtained after cloning and used for bidirectional sequence analyses using M13F and M13R universal primers. Sequencing was performed by Macrogen (Seoul, Korea).

### 2.6. Bioinformatics to Identify Thrips Species and Biotypes

Before sequence analysis, DNA sequences were trimmed and aligned with EditSeq and ClustalW program of MegaAlign (DNASTAR, Version 6.0, Madison, WI, USA), respectively. According to Kim et al. [13], phylogenetic trees were generated by the Neighbor-Joining method using a software package of MEGA6.06 (www.megasoftware.net, accessed on 30 October 2021). Evolutionary distances were computed using the Poisson correction method. Bootstrapping values were obtained with 1000 repetitions to support branching and clustering.

### 2.7. Random Amplified Polymorphic DNA (RAPD) Analysis of Local Populations

In each local population, 30 individuals were randomly chosen for RAPD analysis. gDNA was extracted as described above. Two RAPD primers, N-8017 (5′-TGCTCTGCCC-3′) and N-8041 (5′-ATCGGGTCCG-3′) [18], were used for genotyping of individual thrips. PCR reaction mixture (25 μL) consisted of 2 μL of gDNA, 2.5 μL of 10× PCR buffer, 2.5 μL of dNTP, 2 μL of a single RAPD primer, 1 μL of Taq DNA polymerase (GeneAll), and 15 μL of deionized distilled water. Using a Cycler (Bio-Rad), PCR was run with a preheating step at 94 °C for 5 min followed by 35 cycles of denaturation at 94 °C for 1 min, annealing at 46 °C for 1 min, and polymerization at 72 °C for 1 min. PCR products were separated by electrophoresis using 2 μL of the reaction mixture, 1% agarose gel, and 1× TAE buffer (40 mM Tris, 20 mM acetate, and 1 mM EDTA). Each separated PCR band was regarded as a RAPD gene locus, of which gene frequencies were estimated for each population. Genetic distance among populations was estimated using PROC CLUSTER program [19].

### 2.8. Bioassay on Susceptibility Variation among Field Populations

A fungal pathogen, *Beauveria bassiana*, was purchased from Hannong Farm (Seoul, Korea) and cultured in potato dextrose broth (PDB; Difco, Fisher Scientific Korea, Seoul, Korea). Its spore and conidial mixture were prepared by filtering with four layers of cheese cloth. For bioassay, a fungal suspension (1.17 × 10^6^ conidia/mL) was used. Spinosad was purchased from Dongbang Agro (Seoul, Korea). An insecticide suspension (50 ppm) was used for the bioassay. The insecticide suspension was applied by a spraying method (~500 µL). Randomly ten *T*. *tabaci* adult insects were used for each replication (three replicates). Circular breeding containers (100 × 40 mm, SPL) were used to conduct this experiment. Mortality was taken every 24 h up to five days.

### 2.9. Statistical Analysis

All studies were performed by one-way ANOVA using PROC GLM of SAS program [19]. Means were compared with the least squared difference (LSD) test. In this study, all experiments were performed with three biologically independent replicates. Data are plotted as mean ± standard error using SigmaPlot (Systat Software, San Jose, CA, USA).

## 3. Results

### 3.1. Seasonal Occurrence of T. tabaci in Welsh Onion Field

Welsh onions are usually cultured twice (‘Spring onion’ and ‘Autumn onion’) per year in Korea. Spring onions were transplanted from nursery to field in mid-May and cultured until late August. During this culture period, *T*. *tabaci* was monitored once a week using yellow sticky traps in Andong, Korea (Figure 1). Adults were captured soon after transplantation in mid-May until harvest in August with some low densities after harvest in early September (Figure 1A). Capture number varied with season (F = 12.94; df = 14, 84; *p* < 0.0001). Peak capture was observed in mid-May~early July when the average temperature was 25–28 °C with relative humidity of 60–80%. However, its seasonal occurrence was different among field conditions (F = 3.67; df = 26, 84; *p* < 0.0001). Among three different filed sites, the highest density was found in Welsh onion cultured under an open field condition. The number of total thrips captured from three places during the entire onion culture period was 1079. These were all females based on the presence of ovipositor (Figure 1B).

### 3.2. Thelytokous Reproduction and Its Genetic Characters

The observation that captured thrips in different field conditions were all females suggested that *T*. *tabaci* infesting Welsh onion had a thelytokous biotype (L2). To test this hypothesis, 15 thrips collected from other Welsh onion sites in Korea were assessed for their COI sequences (Figure 2). These sequences were aligned with 32 sequences classified into three biotypes (T, L1, and L2) (Figure 2A). In the resulting phylogeny tree, 32 reference sequences were separated into T biotype (four sequences), L1 biotype (nine sequences), and L2 (19 sequences). In this phylogeny tree, all 15 onion thrips infesting Welsh onion were clustered with sequences classified into L2 biotype. In addition, these 15 samples shared two residues at two nucleotide polymorphic positions (Figure 2B), a characteristic of the L2 biotype.

To test the thelytokous reproduction of *T*. *tabaci*, larvae were individually reared in a laboratory condition and their progeny production from a single parent was assessed (Table 1). These single parents produced all female progeny. Female progeny production was observed even when individually reared adults were randomly selected and allowed to reproduce in a group of 4 or 10 individuals.

### 3.3. Genetic Diversity of Thelytokous Local Populations

Thelytokous parthenogenesis might limit genetic diversity due to the lack of mating. To measure genetic diversity of thrips infesting Welsh onion, *T*. *tabaci* was collected from six different localities (Figure 3A). Using RAPD, sequence diversity was compared for these local populations (Figure 3B). Two RAPD markers (8017 and 8041) produced 5–7 PCR products representing loci. They were then used for genotyping of individuals from local populations for clustering analysis. RAPD 8041 marker separated these populations by 1.2 genetic diversity while RAPD 8017 marker did so by 0.95 genetic diversity. Surprisingly, two Andong subpopulations (‘Yongsang’ and ‘Songcheon’) were not clustered with either RAPD marker. Two remote populations (‘Andong and Yangsan’ or ‘Andong and Gampo’) were closely related based on RAPD analyses.

### 3.4. Variation in Insecticide Susceptibility of Local Populations—No Directional Population Subdivision

To explain the genetic diversity of *T*. *tabaci* in apparent phenotype, these local populations were compared for their insecticide susceptibility (Figure 4). Different populations showed differential susceptibilities to a chemical insecticide, Spinosad (F = 6.85; df = 7, 10; *p* = 0.0037) and a microbial insecticide, *B*. *bassiana* (F = 9.03; df = 7, 10; *p* = 0.0012). However, local variation appeared to be independent of geographical distance as two Andong populations exhibited a significant difference in susceptibility to *B*. *bassiana*.

## 4. Discussion

This study analyzed reproductive modes and genetic variation for field populations of *T*. *tabaci*. These field populations were obtained from thrips infesting Welsh onion. To analyze the reproductive mode, COI sequence analysis and laboratory mating assay were performed. RAPD was used to investigate genetic variation of field populations. All these analyses indicate that *T*. *tabaci* infesting Welsh onion in Korea has a thelytokous type of parthenogenesis.

Thelytokous parthenogenesis was observed for field-collected adults using sticky traps. The monitoring was performed in three fields cultivating Welsh onion during May–August. A main peak occurred in June. However, population sizes were different among field conditions. Specifically, more thrips were collected by traps in an open field condition than in a greenhouse condition. This suggests that thrips frequently migrate in field conditions, consistent with a previous report by Kim et al. [13]. Considering the relative proximity (1–2 km apart) of monitoring places, greenhouse conditions might hinder thrips’ movement, which results in lower captures under greenhouse conditions. All thrips collected by traps during the entire onion culture period were females. To confirm this unisex finding, field populations were individually reared in the laboratory. It was found that they only produced female progeny. These findings support that these thrips populations have a thelytokous type of parthenogenesis.

In recent years, the study of the mitochondrial regions of DNA including COI, using molecular biology, has proved useful in understanding some biological aspects of onion thrips populations [20]. The phylogenetic analysis of COI sequences supported the thelytokous reproductive mode of *T*. *tabaci* infesting Welsh onion. The issue of cryptic species of *T*. *tabaci* was initially raised by Zawirska [8], who suggested that *T*. *tabaci* had the following two biotypes: a ‘tabaci type’ and a ‘non-tobacco (called communis) type’. Later, host-plant transfer experiments and field surveys support the cryptic species hypothesis that *T*. *tabaci* is a heterogeneous taxon [21]. With the spread of use of molecular techniques in population genetics, [9] have clarified this issue using COI sequences (400 bp). They found the following three biotypes: one tobacco (T) biotype and two leek (L1 and L2) biotypes. Their population subdivision index (FST) values ranged from 0.824 to 0.954 and their divergence time was predicted to be 28–21 million years ago. Our current phylogenetic analysis results support these three biotypes as all populations collected from Welsh onion are classified as L2 biotype characterized by asexual thelytokous parthenogenesis, unlike sexual arrhenotokous parthenogenesis of L1 biotype. It is currently unknown how thrips can change between L1 and L2 biotypes of reproductive mode. Although there is no such information on thrips, a social insect (*Apis mellifera capensis*) is known to exhibit alternative thelytokous reproduction to prepare pseudoqueens [22], in which a gene thelytoky is expressed to mediate the unique reproductive mode. An orthologous or alternative gene(s) in *T*. *tabaci* remains to be identified to explain such reproductive mode change.

RAPD analysis indicated the existence of genetic diversity in six local populations of *T*. *tabaci*. RAPD is a type of polymerase chain reaction (PCR) with a short primer (8–12 nucleotides) that can randomly amplify products depending on target genome variation [23]. Our current study used 10-mer primers and produced 5 and 7 PCR products, respectively, in which the product number varied among individuals. Both RAPD primers did not the identical clustering pattern of the populations. However, both RAPD clustering patterns indicate that the genetic variation among local populations did not follow the local distance of populations. For example, two Andong populations were not clustered, but separately clustered with relatively long-distance populations. The genetic diversity was further supported by variations in insecticide susceptibility of these populations. Two Andong populations were different in the susceptibility test. These results suggest that its genetic variation might not have arisen from the chance of interbreeding, supporting the thelytokous type of parthenogenesis. In China, the genetic diversity of *T*. *tabaci* has also been assessed using COI sequences of 12 local populations [24]. A total of six haplotypes were found, all of which were thelytokous. Although these populations had different sequences, there was no correlation between genetic distance and geographical distances [24]. Thus, population subdivision of thelytokous parthenogenesis might have been induced by random genetic drift (RGD) among different evolutionary forces. RGD is also called Sewall Wright effect. It usually occurs in small and isolated populations with genetic islands due to random samplings derived from a population [25]. Reproductive isolation due to asexual thelytokous parthenogenesis may consider the progeny from a single female to be a genetic island from a genetic pool of an entire population. Thus, a single female progeny might be subjected to population subdivision due to RGD. However, little is known on the original population of these subpopulations. Thus, it is hard to conclude the population variation of *T. tabaci* due to RGD. Rather, the variation may be arisen from reproductive isolation, which prevents gene flow among subpopulations. In addition, *T*. *tabaci* may maintain genetic diversity by polyploidy as seen in variation in competency to transmit tomato spotted wilt virus (TSWV) depending on different modes of reproduction [26]. Tetraploidy of *T*. *tabaci* was detected using microsatellite DNA and might be arisen from apomictic parthenogenesis of the thelytokous reproductive mode to possess more alleles than diploidy [26].

In summary, this study reports that *T*. *tabaci* infesting Welsh onion can be classified into L2 biotype characterized by thelytokous parthenogenesis, and that onion thrips populations have genetic variations probably arisen from reproductive isolation and polyploidy.

## Figures and Tables

**Figure 1 insects-13-00078-f001:**
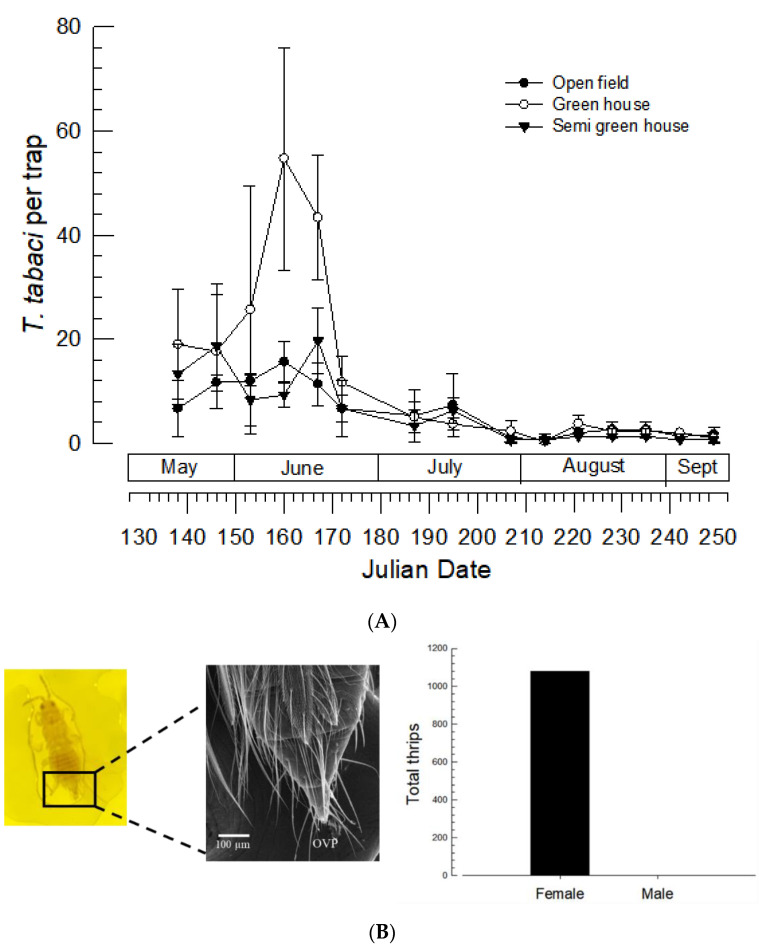
Seasonal occurrence of *T*. *tabaci* in Welsh onion fields. Onions were transplanted in early May. Monitoring traps were installed on 20 May 2021 using yellow sticky plates, and the number of captures was counted every week. Three traps were placed at each field. Onions were harvested at the end of August. (**A**) Occurrence of onion thrips, *T*. *tabaci*, per trap in three different fields in Andong, Korea. Three fields were separated by more than 1 Km. Error bar indicates standard deviation. (**B**) Biased sex ratio of onion thrips collected from fields. Ovipositor (‘OVP’) was used to discriminate females. Left panel shows both stereomicroscope (50×) and SEM (5000×) photos. Right panel indicate sex analysis of total collections from three fields.

**Figure 2 insects-13-00078-f002:**
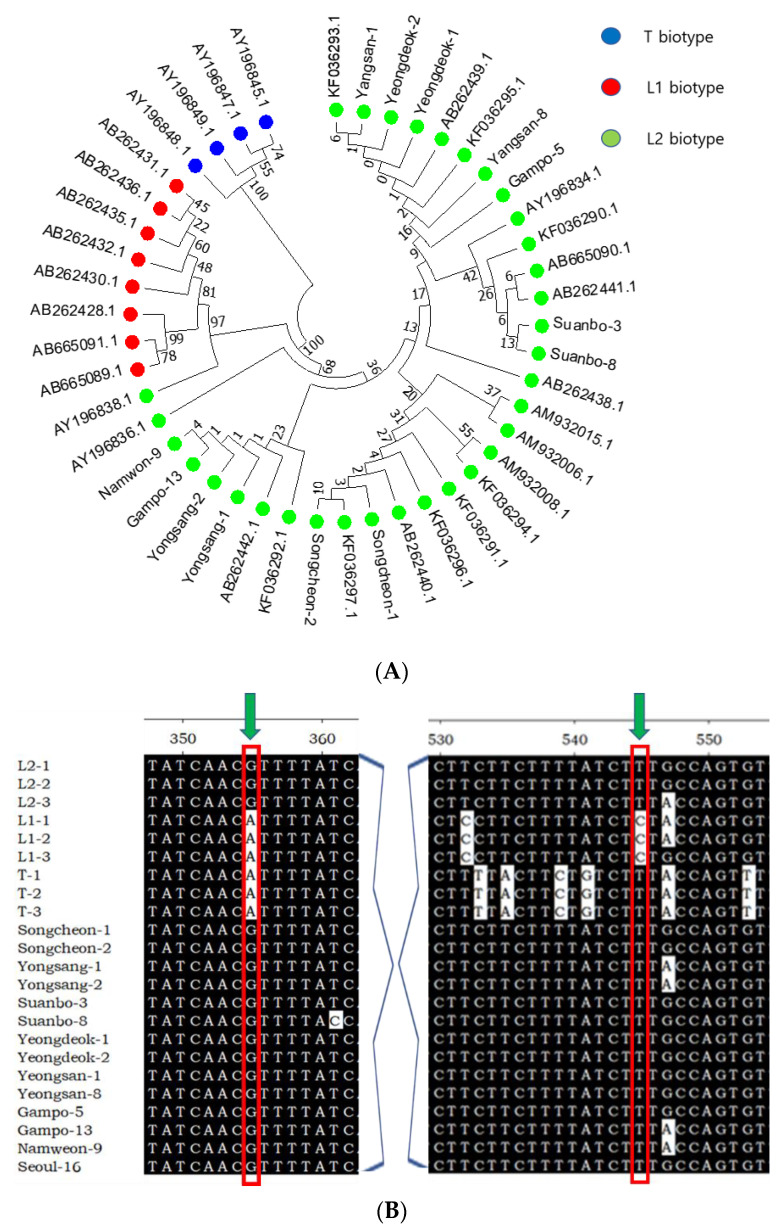
Biotype of *T*. *tabaci* infesting Welsh onion in Korea. (**A**) Phylogeny tree analysis of 12 individuals from different local populations along with known biotypes: tobacco type (T biotype), arrhenotokous type (L1 biotype), and thelytokous type (L2 Biotype). The tree was generated by the Neighbor Joining tree method using the software package MEGA6.0. The tree is drawn in evolutionary distances computed using the Poison correction method. Bootstrap values on nodes were obtained by 1000 repetitions. Individuals are denoted with GenBank accession numbers. (**B**) Conserved nucleotide sequences (red rectangle) that are characteristics of the L2 biotype.

**Figure 3 insects-13-00078-f003:**
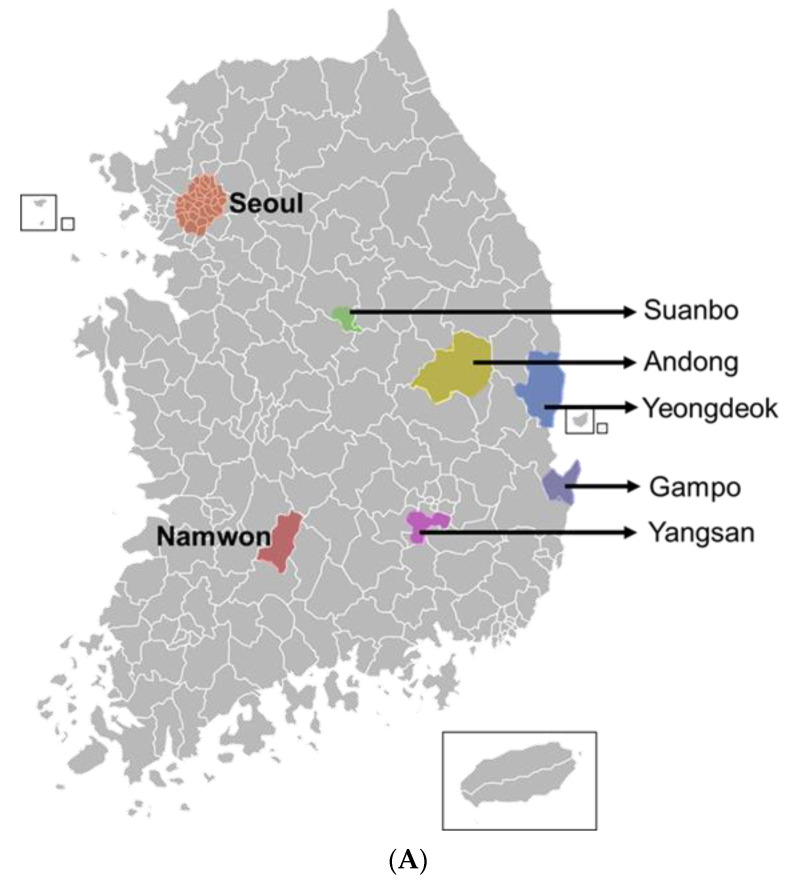
Genetic distance among local populations of *T*. *tabaci* infesting Welsh onion using RAPD. (**A**) A map indicating sampling sites (indicated by arrows) of *T*. *tabaci* in Korea. (**B**) Clustering analysis of RAPD polymorphic markers (8017 and 8041) using PROC CLUSTER [19].

**Figure 4 insects-13-00078-f004:**
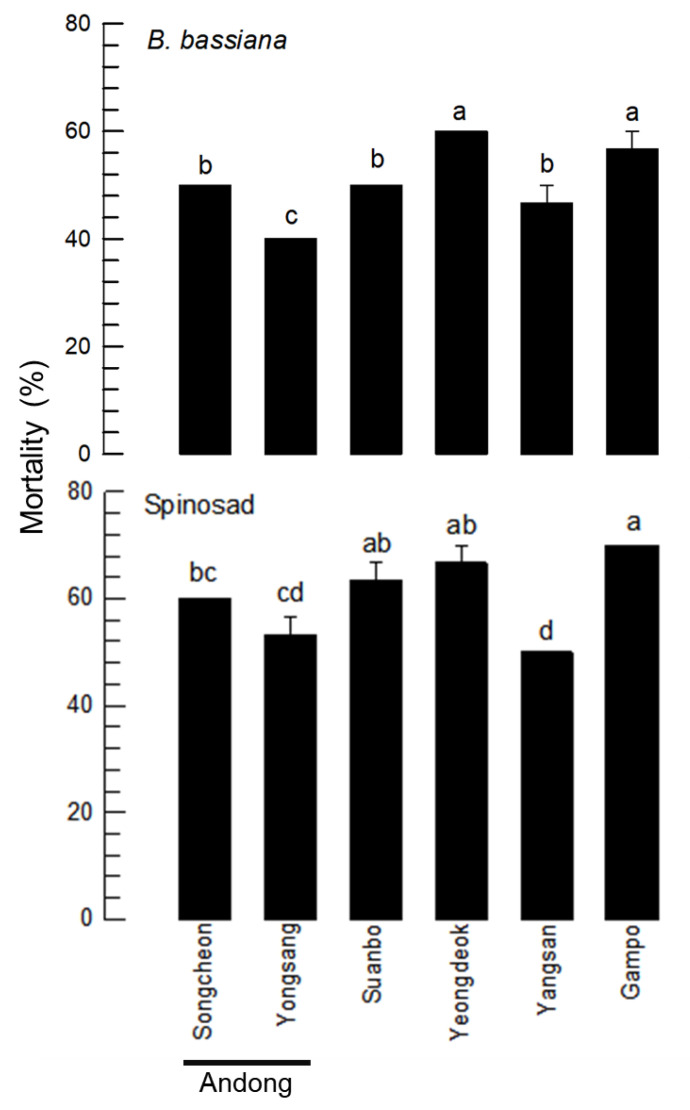
Variation in insecticide susceptibility of *T*. *tabaci* infesting Welsh onion. *Beauveria bassiana* (‘Bb’) and Spinosad at 1.17 × 10^6^ conidia/mL and 50 ppm, respectively, were sprayed to assess thrips’ susceptibility to them. Insecticidal activities of Bb and Spinosad were evaluated at 5 days and 4 days after treatment, respectively. Each treatment used 10 adults and replicated three times. Different letters above means were different using among means at Type I error = 0.05 (LSD test).

**Table 1 insects-13-00078-t001:** Progeny test using different numbers of mating parents.

Mating Parents	Replication	Alive Adults	Total Progeny	Progeny/Parent	
				Female	Male
1	30	13	37	5.4 ± 0.9	0
4	10	26	96	3.6 ± 0.3	0
10	5	25	75	3.1 ± 0.2	0

## Data Availability

Not applicable.

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
