# Peer review of "Thelytokous Reproduction of Onion Thrips, Thrips tabaci Lindeman 1889, Infesting Welsh Onion and Genetic Variation among Their Subpopulations"

_insects, 2022, doi:10.3390/insects13010078_

Round 1

Reviewer 1 Report

Firstly, The paper titled “ Genetic Variation in Thelytokous Populations of Onion Thrips, Thrips tabaci, Infesting Welsh Onion” was  assessed by COI gene and RAPD method.  Local thrips populations exhibited significant variations in susceptibility to chemical and biological insecticides. Thus, population subdivision of thelytokous parthenogenesis might have been induced by random genetic drift (RGD) among different evolutionary forces.
This result should be carefully because the different genetic population were not checked the origin. If all populations are origin from one population the deduce can be explained. However, if all populations are origin from different populations, the different genetic populations can not be explained by RGD. So, at present, the authors did not know the orgion of different pupulaions. The authors should be carefully given the results.

Secondly, the title is not including all works. It should be considered to give suitable title.

It is better to add nominated person and time. " Lindeman 1889" after the species name.

Other minor revisions:

  1. How to eliminate the pseudogenes? in mehtods about COI gene.
  2. Why did not the authors sequence the PCR production directly? If the authors used the Cloning sequence, one individuals of Thrips tabaci should be sequenced at least three clonings to eliminate the affection of Nuclear pseudogenes.
  3. How about the genetic distances among three clades?  These sequences were aligned with 32 sequences classified into three biotypes (T, L1, and L2) (Fig. 2A). In the resulting phylogeny
    tree, 32 reference sequences were separated into T biotype (four sequences), L1 biotype (nine sequences), and L2 (19 sequences).
  4. Check " Two RAPD primers, N-8017 (5′-TGCTCTGCCC-3′) and N-8041 (5′-ATCGGGTCCG-3′), were used for genotyping of individual thrips. " Two RAPD primers are right or not? Is the primers first reported in this paper or cited by reference? if cited, add reference.
  5. How to explain the different phylogenetic relationship of  populations using two RAPD primers?
  6.  Figure 4. is not standard drawing. Redraw it.
  7. L293, Add authors.

Author Response

Comment #1-1: Firstly, The paper titled “Genetic Variation in Thelytokous Populations of Onion Thrips, Thrips tabaci, Infesting Welsh Onion” was assessed by COI gene and RAPD method.  Local thrips populations exhibited significant variations in susceptibility to chemical and biological insecticides. Thus, population subdivision of thelytokous parthenogenesis might have been induced by random genetic drift (RGD) among different evolutionary forces.

This result should be carefully because the different genetic population were not checked the origin. If all populations are origin from one population the deduce can be explained. However, if all populations are origin from different populations, the different genetic populations can not be explained by RGD. So, at present, the authors did not know the orgion of different pupulaions. The authors should be carefully given the results.

Response: This comment is highly informative. Our conclusion of the population subdivision should be modified. First, we delete RGD from this manuscript. Second, the related areas are corrected as population isolation as follows:

In Abstract: “These results suggest that genetic variations of T. tabaci are arisen from population subdivision due to asexual thelytokous reproductive mode.”

In Discussion: “Thus, a single female progeny might be subjected to population subdivision due to RGD. However, little is known on the original population of these subpopulations. Thus it is hard to conclude the population variation of T. tabaci due to RGD. Rather, the variation may be arisen from reproductive isolation, which prevents gene flow among subpopulations. In addition, T. tabaci may maintain genetic diversity by poly-ploidy as seen in variation in competency to transmit tomato spotted wilt virus (TSWV) depending on different modes of reproduction [25]. Tetraploidy of T. tabaci was detected using microsatellite DNA and might be arisen from apomictic parthenogenesis of the thelytokous reproductive mode to possess more alleles than diploidy [25].

  In summary, this study reports that T. tabaci infesting Welsh onion can be classified into L2 biotype characterized by thelytokous parthenogenesis and that onion thrips populations have genetic variations probably arisen from reproductive isolation and polyploidy.”

Comment #1-2: The title is not including all works. It should be considered to give suitable title.

Response: Title is modified by clarifying subdivision of populations as follows: “Thelytokous Reproduction of Onion Thrips, Thrips tabaci, In-festing Welsh Onion and Genetic Variation among Their Sub-populations”

 Comment #1-3: It is better to add nominated person and time. " Lindeman 1889" after the species name.

Response: Added at title and abstract. Also the first citation in text.

 Question #1-4: How to eliminate the pseudogenes? in mehtods about COI gene. Why did not the authors sequence the PCR production directly? If the authors used the Cloning sequence, one individuals of Thrips tabaci should be sequenced at least three clonings to eliminate the affection of Nuclear pseudogenes.

Response: COI sequence was only 485 bp and sequenced in a bi-directional way. These two sequences in each sample were perfectly matched. Thus we did not need to edit the sequences any more.

Question #1-5: How about the genetic distances among three clades?  These sequences were aligned with 32 sequences classified into three biotypes (T, L1, and L2) (Fig. 2A). In the resulting phylogeny tree, 32 reference sequences were separated into T biotype (four sequences), L1 biotype (nine sequences), and L2 (19 sequences).

 Response: Instead calculating genetic distance, we calculated Bootstrap values to support the branching and clusterings. The parsimony was calculated by 1,000 repetitions.

Question #1-6: Check " Two RAPD primers, N-8017 (5′-TGCTCTGCCC-3′) and N-8041 (5′-ATCGGGTCCG-3′), were used for genotyping of individual thrips. " Two RAPD primers are right or not? Is the primers first reported in this paper or cited by reference? if cited, add reference.

Response: Reference added as follows: “16.         Christopher, S.; Francesco, F.; Andrew, B.; Bernie, C.; Hong, L.; Paul, F. Evolution, weighting, and phylogenetic utility of mitochondrial gene sequences and a compilation of conserved polymerase chain reaction primers. Ann Entomol Soc Am. 1994, 87, 651-701.”.

Question #1-7: How to explain the different phylogenetic relationship of populations using two RAPD primers?

Response: We interpreted the discrepancy with respect to non-directional subdivision of these populations. Two RAPD primers commonly separate the populations without any distance-based subdivision of the populations. We add this additional explanation to Discussion as follows: “Both RAPD primers did not the identical clustering pattern of the polulations. Howev-er, both RAPD clustering patterns indicate that the genetic variation among local pop-ulations did not follow the local distance of populations.”

Comment #1-8: Figure 4. is not standard drawing. Redraw it.

Response: We draw this figure by combining two graphs to concisely illustrate the variation among populations in both insecticides. The originals were as follows. Please attach file.

 Comment #1-9: L293, Add authors.

 Response: Added

Reviewer 2 Report

Dear Authors,
I carrefully read the submitted manuscript titled "Genetic Variation in Thelytokous Populations of Onion Thrips, Thrips tabaci, Infesting Welsh Onion".

The aspects investigated are very interesting, however, in all honesty, I found many critical aspects in your manuscript. The work needs extensive editing before it can be accepted for publication.

Here are my main suggestions:
The keywords "Welsh onion and Thrips tabaci" are already included in the title and hence do not fulfil the purpose of keywords, i.e. increasing indexing efficiency and visibility. They should be replaced by unique keywords not included in the title.

Line 38: Please replace "Allium fistulosum" with "Allium fistulosum L.";

Lines 42- 43. Please capitalize the initials of the virus and add its acronym in brackets (IYSV). Please replace "tospovirus" with "Tospovirus";

Lines 44-45: The sentence" Onion thrips is a cosmopolitan species with a wide host range, allowing it to have genetic diversity due to different hosts in various local regions" requires a reference.

Line 50. Replace "....suggested two distinct biotypes." With "....suggested two distinct biotypes within the thrips populations."

Line 60: Please, replace "T. thrips" with "T. tabaci".

Line 67: Please specify how the thrips were collected in the field. The protocol used should be mentioned. I provide you with a recent document that may help you in this objective and which I suggest you use as a reference: http://www.bulletinofinsectology.org/pdfarticles/vol74-2021-241-251marullo.pdf

Line 68: Please move Fig. 3a to this paragraph. In addition, I suggest the authors provide a table with the geographical coordinates of the areas where sampling/monitoring was carried out.

Lines 69-75: Please mention the protocol used for thrips rearing in the lab.

Line 70: Please replace "Allium fistulosum" with "A. fistulosum".

Lines 83-85. This sentence is extremely ambiguous. Thrips are very difficult insects to identify morphologically. Furthermore, it is almost impossible to identify specimens caught with sticky traps with the naked eye. Generally, morphological identification requires the preparation of slides and the use of dichotomous keys (which the authors do not mention!). I strongly recommend that the authors revise the whole paragraph or provide further clarification.

Lines 89-90: Please explain this process better.

Line: Please specify from how many samples the DNA was extracted. Please clarify whether the DNA was extracted from field samples or those bred in the laboratory. There is a lot of confusion.

Lines 102-104: Please give names of primers used and reference literature!

Lines 114-120. How many sequences were obtained? Please specify. Authors need to found the evolutionary model to compute the phylogenetic analysis (with some specific software as Modeltest or Jmodeltest); I would suggest to use ML or Bayesian inference also instead of NJ solely to support the phylogenetic relationship of studied specimens, I found branches not well-supported. Here you will find a paper that addresses some common aspects of your manuscript:  https://doi.org/10.3390/insects11080489

Line 149: Please report the reference for SigmaPlot software.

The Materials and Methods section is not well laid out, and this does not allow a clear understanding of the different steps performed in the study. Many sentences in the Results should be moved to the Materials and Methods section. Similarly, much information reported in the discussion should be moved to the results section. The results should also be clear and well organized.

The discussion should be completely rewritten after reorganizing the previous sections of the manuscript.

The bibliographical references are not sufficient to support the results and hypotheses formulated by the authors, please improve them significantly.

Author Response

Comment #2-1: The keywords "Welsh onion and Thrips tabaci" are already included in the title and hence do not fulfil the purpose of keywords, i.e. increasing indexing efficiency and visibility. They should be replaced by unique keywords not included in the title.

 Response: Replaced with new key words “RAPD and population”

 Comment #2-2: Please replace "Allium fistulosum" with "Allium fistulosum L."

 ResponseReplaced   

Comment #2-3: Please capitalize the initials of the virus and add its acronym in brackets (IYSV). Please replace "tospovirus" with "Tospovirus"

 Response: Added the acronym (IYSV) and corrected with “Tospovirus” 

Comment #2-4: The sentence" Onion thrips is a cosmopolitan species with a wide host range, allowing it to have genetic diversity due to different hosts in various local regions" requires a reference.

Response: Reference is added as follows: “6.       Diaz-Montano, J.; Fuchs, M.; Nault, B.A.; Fail, J.; Shelton, A.M. Onion thrips (Thysanoptera: Thripidae): a global pest of in-creasing concern in onion. J. Econ. Entomol. 2011. 104, 1-13.”

Comment #2-5: Replace "....suggested two distinct biotypes." With "....suggested two distinct biotypes within the thrips populations."

Response: Replaced

Comment #2-6: Please, replace "T. thrips" with "T. tabaci".

Response: Replaced

Comment #2-7: Please specify how the thrips were collected in the field. The protocol used should be mentioned. I provide you with a recent document that may help you in this objective and which I suggest you use as a reference: http://www.bulletinofinsectology.org/pdfarticles/vol74-2021-241-251marullo.pdf

Response: We add the information as follows: “Soon after transplanting onions from the nursery to fields in early May, yellow sticky traps (15 × 25 cm, Green Agrotech, Gyeongsan, Korea) were installed with three repli-cations in each field because sticky traps and cards are used in both open fields and greenhouses for monitoring the early occurrence and change in population size during different seasons [15].”

  1. Marullu, R.; Bonsignore, C.P.; Vono, G. Thrips: a review of sampling methods in relation to their habitats. Bull. Insectol. 2021, 74, 241-251.

Comment #2-8: Please move Fig. 3a to this paragraph. In addition, I suggest the authors provide a table with the geographical coordinates of the areas where sampling/monitoring was carried out.

Response: The information is added as follows: “Seven different places (see Fig. 3A) include Seoul (37o36’12’’N, 127o8’37’’E), Suanbo (36o50’54’’N, 127o59’28’’E), Andong (36o33’33’’N, 128o43’44’’E), Yeongdeok (36o24’36’’N, 129o22’30’’E), Gampo (35o80’28’’N, 129o50’28’’E), Yangsan (35o41’46’’N, 129o05’90’’E), and Namwon (35o40’57’’N, 127o36’61’’E).”

Comment #2-9: Please mention the protocol used for thrips rearing in the lab.

Response: Reference is added as follows: “Larvae and adults of T. tabaci were collected from six different regions (See Fig. 3A). Rearing conditions were temperature, 25 ± 1°C; photoperiod, 16 h/8 h of light/dark; and relative humidity, 70 ± 5% according to Reiter et al [14].”

  1. Reiter, D.; Péter, F.; Sojnóczki, K.; Kristóf, D.K.; József, F. Laboratory rearing of Thrips tabaci Lindeman: a review. Die Boden-kultur 2015, 66, 33-40.

Comment #2-10: Please replace "Allium fistulosum" with "A. fistulosum".

Response: It was mentioned before and so deleted to avoid redundancy.

Comment #2-11: This sentence is extremely ambiguous. Thrips are very difficult insects to identify morphologically. Furthermore, it is almost impossible to identify specimens caught with sticky traps with the naked eye. Generally, morphological identification requires the preparation of slides and the use of dichotomous keys (which the authors do not mention!). I strongly recommend that the authors revise the whole paragraph or provide further clarification. Please explain this process better.

Response: We agree on the concerns raised by reviewer. Thus, we add the following details in the morphological identification: “To avoid any errors in the identification by contamination of Frankliniella occidentalis and F. intonsa, which are common in Korea, we used the morphological characters [16] to diagnose these two species.”

Comment #2-12: Please specify from how many samples the DNA was extracted. Please clarify whether the DNA was extracted from field samples or those bred in the laboratory. There is a lot of confusion.

Response: For COI sequencing, genomic DNA was extracted from around 5-6 adult thrips which was collected from field population of different region. This information is added.

Comment #2-13: Please give names of primers used and reference literature!

Response: Primer names and reference are added.

Comment #2-14: How many sequences were obtained? Please specify. Authors need to found the evolutionary model to compute the phylogenetic analysis (with some specific software as Modeltest or Jmodeltest); I would suggest to use ML or Bayesian inference also instead of NJ solely to support the phylogenetic relationship of studied specimens, I found branches not well-supported. Here you will find a paper that addresses some common aspects of your manuscript:  https://doi.org/10.3390/insects11080489

Response: 32 sequences were collected from NCBI to do the phylogenetic analysis…….

In addition, our NJ clearly separated three host types.

Comment #2-15: Please report the reference for SigmaPlot software.

Response: Reference source is added.

Round 2

Reviewer 2 Report

Dear Authors, I have carefully re-read your manuscript. After the first revision, the work has partially improved, but there are still several parts that need improvement.

Find more suggestions below:

Replace population with thrips population; add another 2/3 keywords.

The subject is very interesting, which is why it deserves a fuller introduction of the different aspects investigated and the methodologies used. Nothing is mentioned the use of DNA Barcoding as a useful tool for this type of studies.

Please add this sentence in the introduction section to justify the use of DNA barcoding as a useful tool for the experiments carried out in this work:

“In recent years, the study of the mitochondrial regions of DNA (Cytochrome c oxidase subunit I - COI), using molecular biology, has proved useful in understanding some biological aspects of onion thrips populations (Marullo et al, 2021)”.

-Marullo, R.; Mercati, F.; Vono, G. DNA Barcoding: A Reliable Method for the Identification of Thrips Species (Thysanoptera, Thripidae) Collected on Sticky Traps in Onion Fields. Insects 202011, 489. https://doi.org/10.3390/insects11080489

2.6. Bioinformatics to Identify Thrips Species and Biotypes: Number the references of the software used and include them in the list of references (DNASTAR ,MEGA etc).

2.8. Bioassay on Susceptibility Variation among Field Populations: Nothing about this test is mentioned in the introduction. Something should be mentioned.

2.9. Statistical Analysis: please consider the same suggestions for paragraph 2.6.

Line 203: Please replace Thrips tabaci with T. tabaci.

Line 267: Please replace T. thrips with T. tabaci.

Figure 3B is unprofessional and in all honesty I think it is not valid for a scientific publication. I suggest removing it from the text. Further information could be provided as Supplementary Material.

I strongly recommend that you improve the Discussion by increasing valid bibliographic references.

I also suggest that you add a (clear and well-articulated) 'Conclusions' section separate from the discussions.

Reference 15, Line 407: Please correct Marullu with Marullo.

Author Response

Comment #1: Replace population with thrips population; add another 2/3 keywords.

Response: Replaced. In addition, we add two more key words as follows: “Welsh onion, Korea”

Comment #2: The subject is very interesting, which is why it deserves a fuller introduction of the different aspects investigated and the methodologies used. Nothing is mentioned the use of DNA Barcoding as a useful tool for this type of studies.

Please add this sentence in the introduction section to justify the use of DNA barcoding as a useful tool for the experiments carried out in this work:

“In recent years, the study of the mitochondrial regions of DNA (Cytochrome c oxidase subunit I - COI), using molecular biology, has proved useful in understanding some biological aspects of onion thrips populations (Marullo et al, 2021)”.

-Marullo, R.; Mercati, F.; Vono, G. DNA Barcoding: A Reliable Method for the Identification of Thrips Species (Thysanoptera, Thripidae) Collected on Sticky Traps in Onion Fields. Insects 202011, 489. https://doi.org/10.3390/insects11080489

Response: It is a highly logical suggestion. We add this information to Discussion. The reference is also added. 

Comment #3: Bioinformatics to Identify Thrips Species and Biotypes: Number the references of the software used and include them in the list of references (DNASTAR, MEGA etc).

Response: We add the reference source as follows: “According to Kim et al. [13], phylogenetic trees were generated by the Neigh-bor-Joining method using a software package of MEGA6.06 (www.megasoftware.net).”

Comment #4: Bioassay on Susceptibility Variation among Field Populations: Nothing about this test is mentioned in the introduction. Something should be mentioned.

Response: Relevant sentence is added to the Introduction as follows: “To test genetic diversity, different local populations of T. tabaci were assessed in their variation in insecticide susceptibility.”

Comment #5: Statistical Analysis: please consider the same suggestions for paragraph 2.6.

Response: Reference [19] indicates the source of stat analysis.

Comment #6: Please, replace "Thrips tabaci" with "T. tabaci".

Response: Replaced.

Comment #7: Please, replace "T. thrips" with "T. tabaci".

Response: Replaced. No more mistakes when we search this error using “T. thrips” in Word.  

Comment #8: Figure 3B is unprofessional and in all honesty I think it is not valid for a scientific publication. I suggest removing it from the text. Further information could be provided as Supplementary Material.

Response: Figure 2B is deleted.

Comment #9: Reference 15, Line 407: Please correct Marullu with Marullo.

Response: Replaced.
